# Entangling single atoms over 33 km telecom fibre

Tim van Leent[1,2,7], Matthias Bock[3,5,7], Florian Fertig[1,2,7], Robert Garthoff[1,2], Sebastian Eppelt[1,2], Yiru Zhou[1,2], Pooja Malik[1,2], Matthias Seubert[1,2], Tobias Bauer[3], Wenjamin Rosenfeld[1,2], Wei Zhang[1,2,6 ✉], Christoph Becher[3 ✉] & Harald Weinfurter[1,2,4 ✉]

Quantum networks promise to provide the infrastructure for many disruptive applications, such as efficient long-distance quantum communication and distributed quantum computing[1,2]. Central to these networks is the ability to distribute entanglement between distant nodes using photonic channels. Initially developed for quantum teleportation[3,4] and loophole-free tests of Bell's inequality[5,6], recently, entanglement distribution has also been achieved over telecom fibres and analysed retrospectively[7,8]. Yet, to fully use entanglement over long-distance quantum network links it is mandatory to know it is available at the nodes before the entangled state decays. Here we demonstrate heralded entanglement between two independently trapped single rubidium atoms generated over fibre links with a length up to 33 km. For this, we generate atom–photon entanglement in two nodes located in buildings 400 m line-of-sight apart and to overcome high-attenuation losses in the fibres convert the photons to telecom wavelength using polarization-preserving quantum frequency conversion[9]. The long fibres guide the photons to a Bell-state measurement setup in which a successful photonic projection measurement heralds the entanglement of the atoms[10]. Our results show the feasibility of entanglement distribution over telecom fibre links useful, for example, for device-independent quantum key distribution[11–13] and quantum repeater protocols. The presented work represents an important step towards the realization of large-scale quantum network links.

Sharing entanglement between distant quantum systems is a crucial ingredient for the realization of future quantum networks. Photons are the tool of choice to mediate entanglement distribution, typically either by controlled light–matter interaction with local memories[14,15], or, as it also will be used here, by entanglement swapping from two pairs of entangled photon-memory states[10,16–18]. Innovative applications of such networks include distributed quantum computing[19] and device-independent quantum key distribution[11–13]. As attenuation losses in the distribution process are inevitable, quantum repeaters will be essential to efficiently distribute entanglement by intermediate nodes.

To minimize absorption along the quantum channel and thus to maximize the distance between neighbouring nodes in quantum networks using the readily available fibre infrastructure, it is necessary to operate at telecom wavelengths. Using quantum frequency conversion (QFC)[20–24], light–matter entanglement distribution at the low loss telecom band has recently been demonstrated for various types of quantum memory, including NV-centres, ions, atoms and atomic ensembles[25–28], even over tens of kilometres of fibre[9,29]. This was mainly enabled by new quantum frequency converters[25,26], which, while preserving the photonic polarization, have reached external device conversion efficiencies as high as 57% (ref. [9]).

For future quantum communication and repeater scenarios, it is vital that the nodes are independent and distant, use long-lived quantum memories and, at the same time, provide the availability of heralded entanglement, that is, there is a well defined signal available that the entanglement distribution succeeded. So far, this has been limited to fibre lengths up to 1.7 km (refs. [5,6]). Recently, great progress was made by demonstrating entanglement between atomic ensembles[7] and multimode solid-state quantum memories[8] using telecom wavelength photons, however, this had limited memory storage times and did not use independent nodes.

Here we report on the distribution of entanglement between two remote quantum nodes—[87]Rb atoms trapped and manipulated independently at locations 400 m apart—generated over fibre links with a length of up to 33 km. The scheme begins with entangling the spin state of an atom with the polarization state of a photon in each node. Subsequently, the photons emitted by the atoms at 780 nm are converted to telecom wavelengths and transferred over several kilometres of fibre to a middle station, where a Bell-state measurement is performed to swap the entanglement to the atoms. We analyse the heralded entanglement between the atoms for different fibre link lengths using correlation measurements of the atom–atom state along three bases. The atoms are

[1]Faculty of Physics, Ludwig-Maximilians-University of Munich, Munich, Germany. [2]Munich Center for Quantum Science and Technology, Munich, Germany. [3]Department of Physics, Saarland University, Saarbrücken, Germany. [4]Max-Planck Institute for Quantum Optics, Garching, Germany. [5]Present address: Institute of Experimental Physics, University of Innsbruck, Innsbruck, Austria. [6]Present address: School of Physics, Xi'An Jiao Tong University, Xi'An, ShannXi, China. [7]These authors contributed equally: Tim van Leent, Matthias Bock, Florian Fertig. ✉e-mail: wei.zhang@xjtu.edu.cn; christoph.becher@physik.uni-saarland.de; h.w@lmu.de

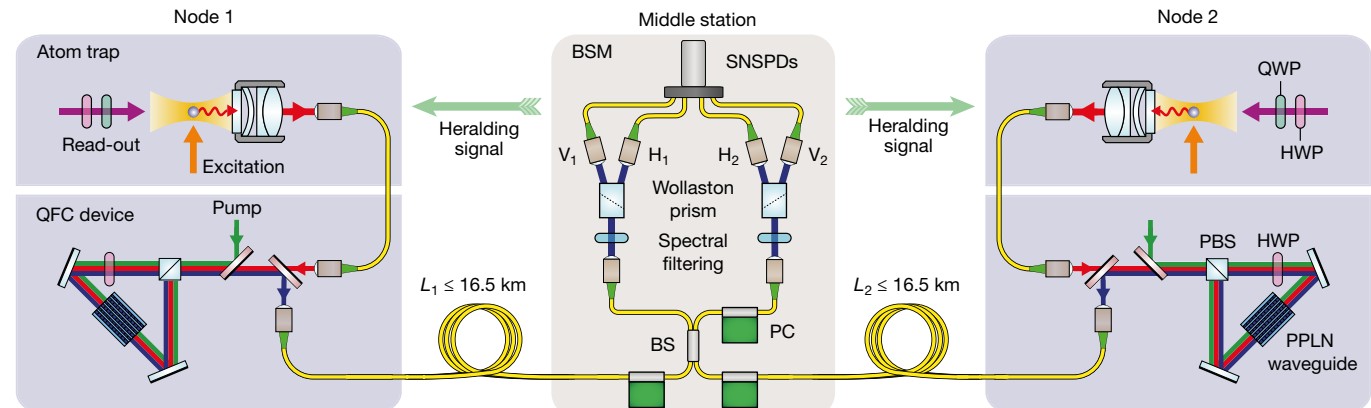

**Fig. 1 | Schematic of the experimental setup.** In each node, located in buildings 400 m apart, a single ⁸⁷Rb atom is loaded in an optical dipole trap. Both atoms are synchronously excited to the state $5^2P_{3/2}|F'=0, m_{F'}=0\rangle$ to generate atom–photon entanglement in the subsequent spontaneous decay. The single photons emitted at a wavelength of 780 nm are collected using high-numerical aperture objectives and coupled into single-mode fibres leading to the QFC devices. There, they are converted to telecom wavelength ($\lambda = 1{,}517$ nm) by difference frequency generation in a periodically poled lithium niobate (PPLN) waveguide located in a Sagnac interferometer type setup, such a configuration fully maintains the polarization quantum state of the photon. The converted photons are guided to a middle station by fibre links with lengths up to 16.5 km, where the entanglement is swapped to the atoms by a BSM. After successfully generating atom–atom entanglement, the atoms are analysed independently by a readout pulse of which the polarization, set by a half-wave plate (HWP) and quarter-wave plate (QWP), defines the measurement setting. PC, polarization controllers.

analysed after a delay corresponding to the two-way communication time to the middle station over the full fibre link length to realistically evaluate the performance for long fibre lengths.

## Quantum network link

Our experiment consists of two similar, independent nodes and a middle station, all of which are located in different laboratories, as illustrated in Fig. 1. The shortest possible fibre connection from node 1 (node 2) to the middle station equals 50 m (750 m), longer fibre links are realized by inserting extra fibres on spools. The fibre length to the middle station is denoted as $L_1$ ($L_2$), with the total link length $L = L_1 + L_2$. In both nodes, a single, optically trapped ⁸⁷Rb atom acts as a quantum memory[30], in which a qubit is encoded in the Zeeman substates of the $5^2S_{1/2}|F=1, m_F=\pm1\rangle$ ground state, with $m_F = +1$ and $m_F = -1$ further denoted as $|\uparrow\rangle_z$ and $|\downarrow\rangle_z$, respectively.

The experimental sequence starts by generating atom–photon entanglement in each node[31]. For this purpose, the atoms are prepared in the initial state $5^2S_{1/2}|F=1, m_F=0\rangle$ by optical pumping and excited to the state $5^2P_{3/2}|F'=0, m_{F'}=0\rangle$. During the spontaneous decay back to the ground state, the atomic spin state becomes entangled with the polarization of the respective emitted photon at 780 nm due to the conservation of angular momentum. This results in the entangled atom–photon state $|\Psi\rangle_{AP} = 1/\sqrt{2}(|\downarrow\rangle_z|L\rangle + |\uparrow\rangle_z|R\rangle) = 1/\sqrt{2}(|\downarrow\rangle_x|H\rangle + |\uparrow\rangle_x|V\rangle)$, where $|L\rangle$ and $|R\rangle$ denote left- and right-circular photonic polarization states, $|H\rangle$ and $|V\rangle$ denote horizontal and vertical linear polarizations. A custom made high-numerical aperture objective is used to collect the atomic fluorescence into a single-mode fibre with an efficiency of 1.0% (1.1%) after an excitation attempt in node 1 (node 2).

Photons with a wavelength of 780 nm now would suffer an attenuation by a factor of ten after propagation through 2.5 km of fibre. To overcome such high-attenuation loss, we use polarization-preserving QFC to transform the wavelength of the photons to the telecom S band, in which one expects attenuation by a factor of ten after only about 50 km transmission. The QFC is realized by mixing the photons with a strong pump field at 1,607 nm inside a non-linear waveguide crystal, converting the wavelength to 1,517 nm by difference frequency generation. Various spectral filtering stages, including a narrow band filter cavity (27 MHz full-width at half-maximum), separate the converted single photons from the strong pump field

and the anti-Stokes Raman background originating from this field. In the shortest fibre configuration, this results in a background of roughly 160 and 170 cps registered at the middle station for light from nodes 1 and 2, respectively. Both converters achieve an external device efficiency of 57%. The pump light is conveniently distributed to the nodes using the telecom fibre network and hence ensures indistinguishable frequencies of the single photons after conversion. For more details about the QFC system and an analysis of the atom–photon entanglement distribution at telecom wavelengths, see ref. [9] and Methods.

After frequency conversion, the photons are guided to the middle station with fibres of different lengths where an interferometric Bell-state measurement (BSM) swaps the entanglement to the atoms[6,10]. The fidelity of the BSM, and hence of the entanglement swapping protocol, is determined by the photons temporal, spectral and spatial indistinguishability[32]. This is optimized by different means. First, the photons impinge on a balanced, single-mode fibre beam splitter to guarantee a perfect spatial overlap. Second, the entanglement generation process in the nodes is synchronized to <300 ps, which is much smaller than the coherence time of the photons determined by the lifetime of the excited state of 26.2 ns. And third, polarization drifts in the long fibres are compensated using an automated polarization control[33]. The photons are detected with four superconducting nanowire single-photon detectors (SNSPDs), which all have an efficiency of >85% and a dark-count rate of <65 cps.

The BSM setup used here analyses the photons in the H/V basis and hence heralds the following two Bell states of the atoms

$$|\Psi^\pm\rangle = \frac{1}{\sqrt{2}}(|\uparrow\rangle_x|\downarrow\rangle_x \pm |\downarrow\rangle_x|\uparrow\rangle_x). \tag{1}$$

Two-photon coincidences are triggered within a hard-wired 208 ns long window that sends a heralding signal back to the nodes. The signal is delayed electronically by $t \geq \ell/\frac{2}{3}c$ to simulate the signalling time back to the nodes, where $\frac{2}{3}c$ approximates the speed of light in an optical fibre and $\ell$ equals $L_1$ or $L_2$ for node 1 or 2, respectively. Although lowering the observed final fidelity, this delay is essential to study the performance of the quantum network link in a realistic scenario.

The quantum state of the atomic memories is finally analysed with a state-selective ionization scheme, in which the state selectivity is

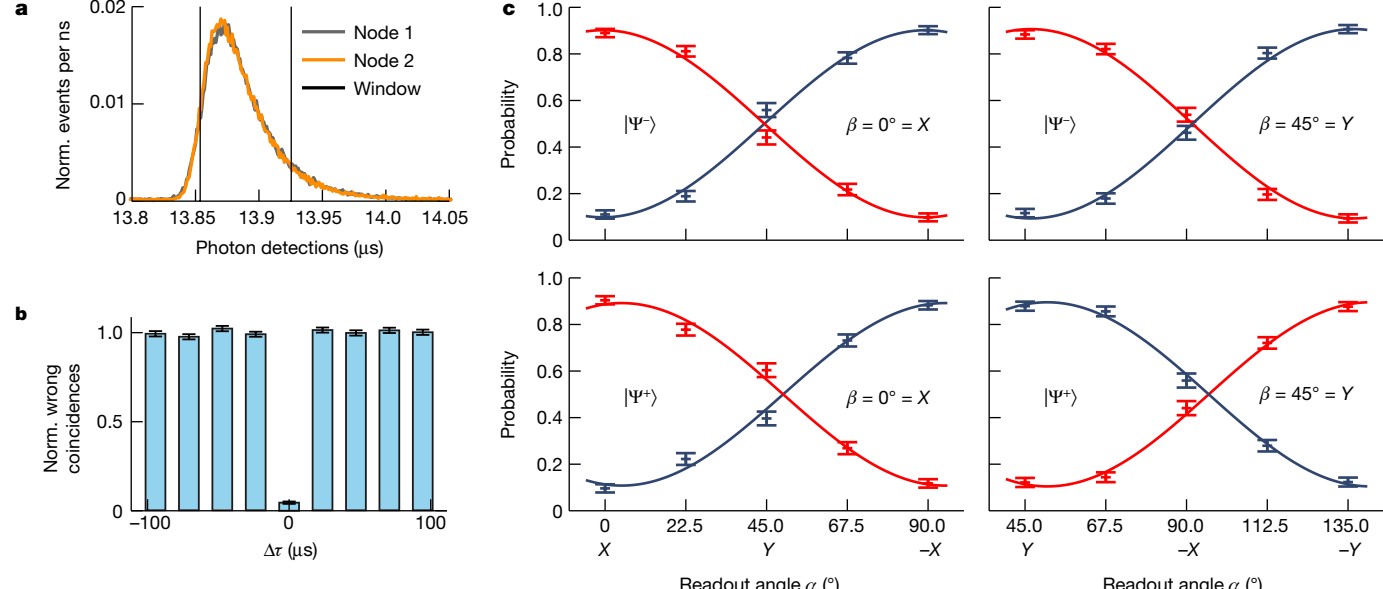

**Fig. 2 | Characterization of the atom–atom entanglement for a fibre length of L = 6 km. a**, Detection time histogram of the photons originating from node 1 and 2 relative to the time of excitation in node 1. For the indicated acceptance window, we observe a SBR of 58 and 65 for nodes 1 and 2, respectively. The temporal overlap of the two photons is >0.98. **b**, Two-photon interference based on the Hong–Ou–Mandel effect. Shown is the normalized (norm.) number of wrong coincidences for various time differences between the two photon wave-packets ($\Delta\tau$). **c**, Atom–atom state correlations for the $|\Psi^-\rangle$ (top) and $|\Psi^+\rangle$ (bottom) states. The correlation probability of the measurement outcome in the nodes is shown in blue, whereas the anticorrelation probabilities are marked in red. The data are fitted with sinusoidal functions resulting in an estimated fidelity for $|\Psi^-\rangle$ and $|\Psi^+\rangle$ relative to a maximally entangled state of 0.826(18) and 0.806(20), respectively. The error bars in this and all subsequent figures indicate statistical errors of one standard deviation.

controlled by the polarization of a readout laser pulse[9]. Both memories have a coherence time $T_2$ of roughly 330 µs, which is achieved by active compensation of external magnetic fields (<0.5 mG) and applying a bias field of tens of milligauss along the $y$ axis. Currently, the coherence time is limited by magnetic field fluctuations along the bias field direction and a position-dependent dephasing originating from longitudinal field components of the strongly focussed dipole trap. For more details and a simulation of the limiting decoherence mechanisms, see Methods.

## Entangling atoms using telecom photons

The evaluation of the entanglement distribution over long fibre links is detailed first for a fibre configuration of $L$ = 6 km. The entanglement generation rate is determined by two factors: the probability of generating an entangled state between the atoms after a synchronized excitation attempt, predominantly reduced by the photon collection efficiency in the nodes, amounts to $\eta = 3.66 \times 10^{-6}$ and the repetition rate of $R$ = 30.8 kHz, which is mainly limited by the communication time between the nodes. This leads to an event rate $r = 1/9$ s$^{-1}$ for coincidence detections in the full hard-wired window. With a duty cycle of the setup of roughly half, including the fraction of time that an atom is present in both traps of 0.60 (5), we observed $N$ = 10,290 entanglement generation events in 54 h. For more details about the entanglement generation rate and duty cycle, see Methods.

Facing Raman background from two QFC devices in addition to detector dark counts, a 70 ns photon acceptance window is applied during the data evaluation, as shown in Fig. 2a. This results in a signal-to-background ratio (SBR) of 58 (65) for detecting a single photon from node 1 (node 2), which is substantially higher than in a previously reported work (ref. [9]) thanks to a more favourable pump-signal frequency combination with respect to the Raman background. For the coincidence detections this leads to a SBR of 48, while accepting roughly 65% of the recorded events.

The quality of the two-photon interference of the converted photons is quantified by the relative occurrence of wrong detector coincidences[10]. These coincidences, that is, $(V_1,V_2)$ and $(H_1,H_2)$, should not occur for perfectly interfering, fully unpolarized photons. For temporally well overlapping photons ($\Delta\tau = 0$), but without background correction, this results in an interference contrast of 0.955(7) (Fig. 2b). Here, two photon emission events from one atom reduce the indistinguishability of the interfering photons by changing the temporal photon shape[34]. By rejecting early detection events, this effect is reduced at the cost of a lower event rate. For more details, see Methods.

To evaluate the atom–atom entanglement we measured the atomic spin states in the two linear bases, $X$ and $Y$. For this, the polarization analysis angle in node 2 was set to $\beta = 0° = X$ and $\beta = 45° = Y$, whereas the analysis angle in node 1 was varied over 90° in steps of 22.5° starting from $\alpha = 0° = X$ and $\alpha = 45° = Y$, respectively. The atom in node 1 (node 2) was analysed at $t_1 = 28.5$ µs ($t_2 = 35.5$ µs) after the respective excitation pulse. The resulting atomic state correlation probabilities $P_{corr} = (N^{\alpha,\beta}_{\uparrow\uparrow} + N^{\alpha,\beta}_{\downarrow\downarrow})/N^{\alpha,\beta}$ and anticorrelation probabilities $P_{acorr} = (N^{\alpha,\beta}_{\uparrow\downarrow} + N^{\alpha,\beta}_{\downarrow\uparrow})/N^{\alpha,\beta}$ are shown in Fig. 2c. The data are fitted with sinusoidal curves giving average visibilities of $\overline{V} = 0.804(20)$ for $|\Psi^-\rangle$ and $\overline{V} = 0.784(23)$ for $|\Psi^+\rangle$. To estimate the state fidelity we need to consider that the third ground state ($5^2S_{1/2}|F = 1, m_F = 0\rangle$) can be populated and hence the atomic states evolve effectively in a 3 × 3 state space. Therefore, a lower bound on the fidelity is given by $\mathcal{F} \geq \frac{1}{9} + \frac{8}{9}\overline{V}$ (Methods), resulting in fidelities of $\mathcal{F} \geq 0.826(18)$ for $|\Psi^-\rangle$ and 0.806(20) for $|\Psi^+\rangle$, relative to a maximally entangled state. (When also accepting the remaining 35% of the coincidences in the hard-wired acceptance window, the average observed fidelity for the states $|\Psi^\pm\rangle$ equals 0.772(10), with a SBR of 22, and an interference contrast of 0.895(8).) The analysis angles chosen allow to evaluate the Clauser–Horne–Shimony–Holt S value[35] for the settings $\alpha = 22.5°$, $\beta = 0°$; $\alpha' = 67.5°$, $\beta = 0°$; $\alpha' = 67.5°$, $\beta = 45°$ and $\alpha'' = 112.5°$, $\beta' = 45°$, whereby $\alpha''$ replaces $\alpha = 22.5°$. This results in an observed value of $S$ = 2.244(63), violating the classical limit of two with 3.9$\sigma$.

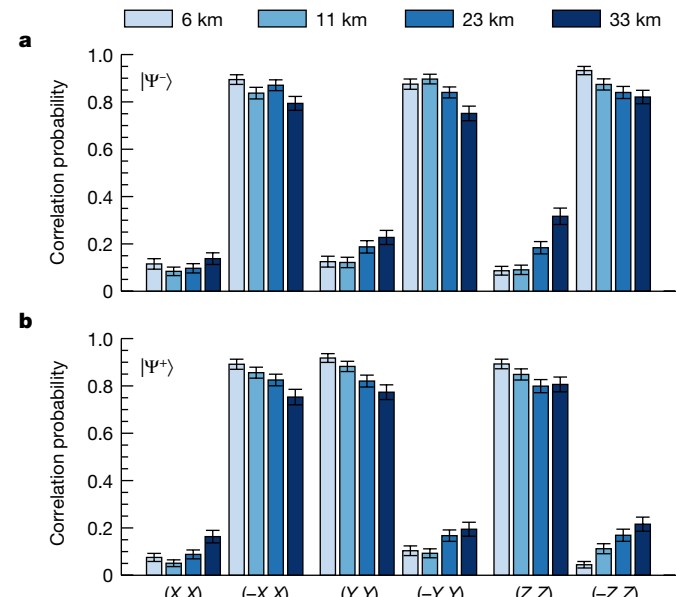

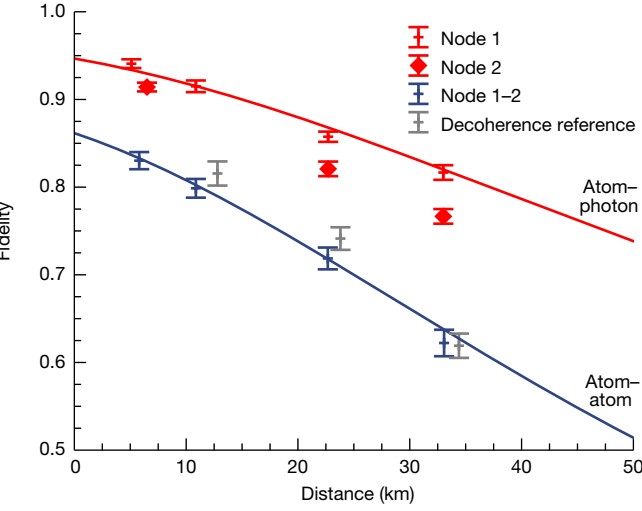

**Fig. 3 | Observation of heralded atom–atom entanglement generated over long fibre links. a,b,** Correlations between the measurement results when analysing the generated atom–atom Bell states $|\Psi^-\rangle$ (**a**) and $|\Psi^+\rangle$ (**b**) for various link lengths. For each link, the states were analysed in three conjugate bases (node 1, node 2), whereby the correlation probability of the measurement result in the nodes equals $P_{corr} = (N_{\uparrow\uparrow} + N_{\downarrow\downarrow})/(N_{\uparrow\uparrow} + N_{\uparrow\downarrow} + N_{\downarrow\uparrow} + N_{\downarrow\downarrow})$. For the different fibre lengths (short to long) 4,281, 4,271, 4,153 and 3,022 entanglement events were recorded within measurement times of 11, 65, 97 and 175 h. Of these events, 62–72% were within the two-photon coincidence acceptance window, resulting in $N = 185$ to 225 events per data point.

**Fig. 4 | Observed entanglement fidelity for various link lengths.** Overview of the observed atom–atom fidelities for different fibre configurations (blue). For completeness, the observed atom–photon fidelities of the states between the nodes and the middle station (red), with $L = 2L_i$, are given (Methods). The solid lines are simulations based on a model taking into account the decoherence of the atomic memories (Methods). The grey points are reference measurements of the atom–atom state decoherence without long fibres but with corresponding readout delay.

## Entangling atoms over up to 33 km of fibre

To determine the quality of entanglement for long fibre links, we performed a series of measurements generating and observing atom–atom entanglement in fibre configurations with a length $L = 6, 11, 23$ and 33 km. For longer links, the event rate reduces because of both the signal attenuation around 0.2 dB km⁻¹ and the longer communication times (Methods). At 33 km, this results in a success probability of $\eta = 1.22 \times 10^{-6}$, a repetition rate of $R = 9.7$ kHz and an event rate of $r = 1/85$ s⁻¹. The entanglement fidelity relative to maximally entangled states was analysed by measurements along three bases ($X$, $Y$, and $Z$). Fig. 3 shows the probability of correlated measurement outcomes in the nodes for each measurement setting combination and fibre configuration.

The fidelity of the observed states is estimated by first determining the contrast in the three measurement bases independently. This is done by taking the absolute difference of the two measured correlation probabilities, $E_k = |P_{k,k} - P_{-k,k}|$ for $k \in \{X, Y, Z\}$, from which the average contrast is computed as $\bar{E} = (E_X + E_Y + E_Z)/3$. When averaging over the observed states $|\Psi^{\pm}\rangle$—which show in our measurement precision similar visibilities—and again assuming the $3 \times 3$ state space, this results in a lower bound on the fidelities $\mathcal{F} = 0.830(10), 0.799(11), 0.719(12)$ and $0.622(15)$ relative to maximally entangled states for $L$ equals 6, 11, 23 and 33 km, respectively. The estimated fidelity for the 6 km fibre configuration is in good agreement with the fidelity estimated from the measurements in two bases presented before (Fig. 2c). Moreover, all observed fidelities clearly exceed the bound of 0.5 and hence clearly witness an entangled state.

The observed fidelities are shown in the dependence of the fibre link length in Fig. 4, where the measured atom–photon entanglement fidelities for states shared between the nodes and the middle station are also shown for comparison (Methods). The atom–atom state

fidelity for different fibre lengths $\mathcal{F}(L)$ is modelled on the basis of simulations of the generation and evolution of the two atom–photon states, which are shown with solid lines. We estimate the visibility of the atom–atom state by the product of the two atom–photon visibilities and the interference contrast of the BSM[36]. Evidently, decoherence of the atomic states dominates the loss in fidelity for longer fibre links. For $L = 33$ km, the states were analysed 171 and 178 μs after excitation in nodes 1 and 2, respectively, which approaches the coherence time of the states. By contrast, the SBR is robust to an increase in fibre length because both the single photons as well as the QFC background are attenuated in the long links. A minor reduction in SBR (42 for $L = 33$ km) is explained by relatively more detector dark counts and can be solved by installing detectors with lower dark counts. Also, polarization drifts are comparably well compensated in all link configurations (Methods).

To verify that the memory decoherence limits the loss in fidelity for long fibre links, we performed a series of measurements without further fibres inserted, however, with the memory readout times electronically delayed according to the long fibre links. The observed fidelities are shown in grey in Fig. 4 at $L = \frac{2}{3}c(t_1 + t_2)/2$ (matching the two-way communication time to the middle station for distance $L$) and show, within the measurement accuracy, no difference in observed fidelity compared to the configuration with long fibres.

## Discussion and outlook

The results clearly indicate the feasibility of turning to large-scale quantum networks and of increasing the line-of-sight separation of the nodes to tens of kilometres by using efficient QFC. To evaluate the performance of entanglement generation in future quantum networks the so called quantum link efficiency was introduced recently[37]. It is defined as the ratio of the entanglement generation rate over the entanglement decay rate and describes how efficiently one can use entanglement as a resource in future quantum networks. Ideally, it should exceed one, that is, entanglement is available on demand as it is generated faster than it decays. However, the link efficiency decays rapidly with length due to both the exponential decrease of the signal detection probability,

but, even more dominantly for the link lengths realized here, also due to the waiting times for classical communication between the nodes (Methods). In this proof-of-principle demonstration, the link efficiency was at the $10^{-5}$ to $10^{-6}$ level, mainly due to the low photon collection efficiency, relatively short coherence times and long fibre links.

To raise the link efficiency, several improvements are feasible: the coherence time of the atomic memories can be increased by the implementation of, first, a new trap geometry to mitigate the position-dependent dephasing and second, a state-transfer to a qubit encoding less sensitive to magnetic fields[38] making possible coherence times in excess of 5 ms, and thus distances of 100 km without notable memory decoherence. On the entanglement generation side, optical cavities could enhance the fluorescence collection efficiency into fibres. For single rubidium atoms, photon collection efficiencies up to 11% have been demonstrated[14]. Yet, even for an almost ideal experimental platform the entangling rate cannot be increased arbitrarily high. To further increase the event rate on the long run it will be mandatory to parallelize the entanglement distribution to regain the scalability. For the neutral atom platform this could be achieved by realizing atom trap arrays[39] and hence potentially increase the entanglement generation rate by orders of magnitude. This concept will also lay the basis to realize quantum repeater network nodes performing entanglement purification and BSMs by local atom–atom gate operations in the array, for instance, using Rydberg interactions[40].

To conclude, using efficient telecom interfaces in our nodes enabled the generation of heralded entanglement between two atomic quantum memories over fibre links with a length as long as 33 km. The results clearly show that improvements on the memory coherence time and entanglement generation rate are mandatory, but will allow to entangle two atomic quantum memories with a fidelity better than 80% over fibres links up to 100 km, thereby paving the way towards long-distance entanglement distribution for future quantum networks.

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

## Methods

### Entanglement generation and single atom state readout schemes

The atom–photon entanglement generation process is visualized in Extended Data Fig. 1a. It starts by preparing the atom in the state $5^2S_{1/2}|F = 1, m_F = 0\rangle$ by optical pumping with an efficiency of 80%. Next, a short laser pulse with 21 ns full-width at half-maximum excites the atom to the state $5^2P_{3/2}|F' = 0, m_{F'} = 0\rangle$. In the subsequent spontaneous decay, the atomic spin state becomes entangled with the polarization state of the emitted photon, denoted as $|\Psi\rangle_{AP} = 1/\sqrt{2}(|\downarrow\rangle_z|L\rangle + |\uparrow\rangle_z|R\rangle) = 1/\sqrt{2}(|\downarrow\rangle_x|H\rangle + |\uparrow\rangle_x|V\rangle)$, where $|H\rangle = i/\sqrt{2}(|L\rangle - |R\rangle), |V\rangle = 1/\sqrt{2}(|L\rangle + |R\rangle), |\downarrow\rangle_x = i/\sqrt{2}(|\uparrow\rangle_z - |\downarrow\rangle_z)$, and $|\uparrow\rangle_x = 1/\sqrt{2}(|\uparrow\rangle_z + |\downarrow\rangle_z)$.

The atomic state readout process is visualized in Extended Data Fig. 1b. After a successful atom–atom entanglement generation event, the atomic spin states of the atoms are individually analysed using a state-selective ionization scheme. This scheme starts by transferring a selected atomic qubit state superposition from the ground state $5^2S_{1/2}|F = 1\rangle$ to the excited state $5^2P_{1/2}|F' = 1\rangle$ with light at 795 nm ('read-out light'). Simultaneously, the excited state is ionized using a bright laser pulse at 473 nm. If the atom possibly decays to the state $5^2S_{1/2}|F = 2\rangle$ before ionization, as indicated with the right-most grey arrow in Extended Data Fig. 1b, a 780-nm cycling pulse will transfer it to the state $5^2P_{3/2}|F' = 3\rangle$, which is also ionized. The state readout is completed by fluorescence collection on a closed atomic transition to check whether the atom is still present in the trap or not. The fidelity of atomic state readout operation is 96%.

The measurement basis of the atomic qubit state is controlled by the polarization of the readout light pulse, which is defined as $\chi = \cos(\alpha)V + e^{-i\phi}\sin(\alpha)H$. Accordingly, two orthogonal atomic qubit state superpositions can be derived of which one is transferred to the excited state by the readout pulse (bright-state) and the other is not (dark state), given as

$$|\Psi\rangle_{Bright-State} = \cos(\alpha)\frac{1}{\sqrt{2}}(|\downarrow\rangle_z - \uparrow\rangle_z) + \sin(\alpha)e^{-i\phi}\frac{i}{\sqrt{2}}(|\downarrow\rangle_z + \uparrow\rangle_z) \quad (2)$$

$$|\Psi\rangle_{Dark-State} = \sin(\alpha)\frac{1}{\sqrt{2}}(|\downarrow\rangle_z - \uparrow\rangle_z) - \cos(\alpha)e^{-i\phi}\frac{i}{\sqrt{2}}(|\downarrow\rangle_z + \uparrow\rangle_z) \quad (3)$$

Note that all states except the dark state are excited and hence ionized, for example, population in the state $5^2S_{1/2}|F = 1, m_F = 0\rangle$ is always ionized. This makes the readout scheme a projection measurement onto the dark state.

An intuitive example of the state selectivity is the case of a $\sigma^+$-polarized readout pulse. In the $z$ basis, as shown in Extended Data Fig. 1b, this pulse will excite an atom in the state $|\downarrow\rangle_z$ to the state $5^2P_{1/2}|F' = 1, m_{F'} = 0\rangle$, however, an atom in the state $|\uparrow\rangle_z$ will not be excited because of the absence of the state $5^2P_{1/2}|F' = 1, m_{F'} = 2\rangle$.

### Atom–photon entanglement distribution at telecom wavelength

The polarization-preserving QFC devices used in this work are described in detail in refs. [9,41]. In contrast to previous work, here a more favourable pump-signal frequency combination is selected with respect to the Raman background: 1,607–1,517 nm instead of 1,600–1,522 nm. This increases the SBR by a factor of four and allows to install a QFC device in both nodes without being limited by the SBR.

Because the quality of the entanglement shared between the two nodes directly depends on the fidelity of the two entangled atom–photon pairs, we individually characterize the atom–photon entanglement generated in both nodes. The generated states are analysed using the same fibre configurations and atomic readout times as during the atom–atom entanglement measurements presented in the main text. For an overview, see Extended Data Table 1. Note that a high fidelity atomic state readout can only be made after a full oscillation period of the atom in the dipole trap, which equals 14.3 and 17.8 μs for nodes 1 and 2, respectively.

The atom–photon entanglement fidelity is analysed following the methods in ref. [9], in which the atomic readout time is now delayed to allow for two-way communication to the middle station for each node over the respective fibre length. The polarization of the photons are measured in two bases, $H/V$ (horizontal/vertical) and $D/A$ (diagonal/antidiagonal), that is, $X$ and $Y$, whereas the atomic analysis angle was rotated over angles including these bases. The atom–photon state correlations are shown for node 1 in Extended Data Fig. 2 and for node 2 in Extended Data Fig. 3.

For the fibre configuration $L = 6$ km, that is, $L_1 = 2.6$ km and $L_2 = 3.3$ km, we find atom–photon state fidelities of 0.941(5) for node 1 and 0.911(6) for node 2, relative to a maximally entangled state, which are mainly limited by the atomic state readout and entanglement generation fidelity. For longer fibre lengths, the fidelity of the entangled state reduces due to magnetic field fluctuations along the bias field direction and the position-dependent dephasing.

### Modelling of the quantum memory decoherence

In both nodes, a single $^{87}$Rb atom is stored in an optical dipole trap where a qubit is encoded into the states $5^2S_{1/2}|F = 1, m_F = \pm 1\rangle$. The dipole trap is operated at $\lambda_{ODT} = 850$ nm with typical trap parameters of, for example, for node 1, a trap depth $U_0 = 2.32$ mK and beam waist $\omega_0 = 2.05$ μm. The qubit evolves effectively in a spin-1 system as the state $5^2S_{1/2}|F = 1, m_F = 0\rangle$ could also be populated. The state fidelity is influenced by two factors: the first one is the a.c. Stark shift originating from the dipole trap and, second, the Zeeman effect arising from magnetic fields.

To model the dephasing of the quantum memories, we simulate the evolution of this spin-1 system while the atom is oscillating in the dipole trap, affected by longitudinal polarization components and external magnetic fields[42]. For this, we first randomly draw a starting position and velocity of an atom from a 3D harmonic oscillator distribution in thermal equilibrium. Second, the motion of the atom is simulated in a realistic Gaussian potential resulting in an atomic trajectory for which the evolution of the atomic state is calculated on the basis of the local optically induced and external magnetic fields. Finally, this is repeated for a large number of trajectories in which the averaged projection for all trajectories yields the simulation result.

The model takes the following independently measured inputs: (1) the trap geometry specified by the beam waist $\omega_0$, which is obtained from knife-edge measurements of the dipole trap beam focus in two dimensions[43]; (2) the trap depth $U_0$, determined by measurements of the transverse trap frequency using parametric heating[44] and the atomic state rephasing period[42] and (3) the atomic temperature $T$, modelled as a Boltzmann distribution that is measured by the release and recapture technique[45]. Inputs 1 and 2 define the position, amplitude and phase of the longitudinal polarization components, whereas inputs 1–3 characterize the atomic trajectories. Furthermore, we include a uniform magnetic field along three directions with shot-to-shot noise following Gaussian distributions.

Extended Data Fig. 4 shows simulation results and measurement data of the state evolution in node 1 for varying state readout orientation and time. The model accurately predicts the evolution of the measured atomic states and shows that the memory storage time is limited by magnetic field fluctuations on the order of <0.5 mG along the bias field direction in addition to the position-dependent dephasing due to the longitudinal field components of the strongly focussed dipole trap. The simulation results presented in the main text consider the envelope of the found oscillating state evolution in three bases.

### Experimental sequence

The entanglement generation sequence is visualized in Extended Data Fig. 5. The sequence starts by trapping an atom in both nodes.

For this, a single atom is loaded from a magneto-optical trap into a tightly focussed dipole trap, which takes roughly 1 s (2 s) for node 1 (2). Every entanglement generation try consist of 3 μs optical pumping (80% efficiency) and an excitation pulse (Gaussian laser pulse with a full-width at half-maximum of 21 ns) to generate atom–photon entanglement in the following decay. Subsequent to each try, a waiting time is implemented to cover the propagation time of the photons in the long fibres. After 40 unsuccessful tries, the atoms are cooled for 350 μs using polarization gradient cooling. The lifetime of the atoms in the trap during the entanglement generation process is 4 s (6 s) for node 1 (2).

To verify whether both traps still store a single atom during the entanglement generation tries, the process is interrupted after 200 ms to check the presence of the atoms. For this, a microelectromechanical systems, a fibre-optic switch is installed in each node at the SM-fibre that is used for the atomic fluorescence collection from the atom trap. The switches guide the atomic fluorescence either to the QFC devices during the entanglement generation tries, or to an avalanche photodiode located at each node during 40 ms of fluorescence collection. Note that the SNSPDs of the BSM cannot be used for this purpose because they are behind narrowband spectral filters.

Regular maintenance tasks lower the duty cycle of the experiment to roughly half for all link lengths. This includes the fraction of time required to simultaneously load an atom in the traps (0.40(5)), the fraction of time used to verify if both traps still store a single atom during the entanglement generation tries (0.18), and the fraction of time used to compensate polarization drifts of the long fibres (0.05).

## Entanglement generation rate

The atom–atom entanglement generation rate $r$ is given by

$$r = \eta R, \tag{4}$$

here $\eta$ equals the success probability for each entanglement generation try and $R$ is the repetition rate of the entanglement generation tries. Both $\eta$ and $R$ are dependent on the link length $L$. In the following, we assume a two-node setup with a middle station halfway the nodes, that is, $L = L_1 + L_2$ with $L_1 = L_2$, where $L_1$ ($L_2$) equals the link length from node 1 (node 2) to the middle station.

The success probability of an entanglement generation try is given by

$$\eta(L) = \eta_{L=0} 10^{-\alpha/10\,L} = \mathcal{O}(exp(-L)), \tag{5}$$

here $\eta_{L=0}$ denotes the success probability for a setup with a zero length link and approximates $5.0 \times 10^{-6}$ for the presented apparatus. This includes the photon collection efficiencies in both nodes after an excitation attempt (1.0 and 1.1%), the transmission of the microelectromechanical system switches (85%*), the efficiency of the frequency conversion devices (57%*), the single-photon transmission efficiencies of the spectral filtering cavities (81%*), the fibre coupling to the single-photon detectors (90%*), the single-photon detector efficiencies (85%*) and the fraction of distinguishable Bell states of 2/4 (*these efficiencies should be included twice). The attenuation rate in optical fibres is denoted by $\alpha$ in units of dB km$^{-1}$, which is reduced using polarization-preserving QFC to telecom wavelength from 4.0 dB km$^{-1}$ at 780 nm to 0.2 dB km$^{-1}$ at 1,517 nm.

The repetition rate of the entanglement generation tries equals

$$R(L) = \frac{1}{T} = \frac{1}{T_{L=0} + \frac{1}{2}L/\left(\frac{2}{3}c\right)} = \mathcal{O}(L^{-1}), \tag{6}$$

where $T$ is the period of an entanglement generation try. The period of an entanglement generation try for a link with $L = 0$ is denoted by $T_{L=0}$ and equals 12 μs for the presented apparatus ($R(0) = 8.3 \times 10^4\,\mathrm{s}^{-1}$). This includes the initial state preparation (3 μs), the entanglement generation (200 ns), and the duration of the polarization gradient

cooling per try (350 μs/40) to counteract the introduced heating during the state preparation and entanglement generation tries. The second term in the denominator gives the communication times between the nodes and the middle station over optical fibres, where $\frac{2}{3}c$ approximates the speed of light in an optical fibre. The factor $\frac{1}{2}$ appears as, in the implemented experimental sequences, the electronic delay for the atomic readout is only applied after a successful heralding event (Methods, Experimental sequence). This is not feasible when the two nodes are physically separated by a distance $L$ and, effectively, reduces the entanglement generation rate by a factor up to two for a physical separation $L$ compared to the values observed in this work.

For $L < 54$ km the entanglement generation rate is mainly reduced by the rapidly decreasing repetition rate $R$, for example, from $L = 0$ to $L = 33$ km, $R(L)$ drops by a factor of 8 while the success probability reduces by a factor 4.5. Only for distances $L > 54$ km will the exponential dependence of the success probability outweigh the dependence of the repetition rate. Extended Data Fig. 6 shows the expected entanglement generation rate according to equations (4)–(6) extrapolated to distances up to 100 km, together with the entanglement generation rate of the data presented in the main text.

Note that the fidelities presented in the main text equal the expected fidelities for a physical separation of the nodes by a distance $L$. Moreover, the measurement times reported in the main text include a duty cycle of the experiment of roughly half, for example, the time effectively used for the entanglement generation tries approximates half of the reported measurement times (Methods, Experimental sequence).

## Atom–atom state fidelity

The state of the atomic quantum memories is encoded in two magnetic sublevels of the rubidium ground state $5^2S_{1/2}|F=1\rangle$, which, however, is a spin-1 system. Besides the qubit states $|m_F = \pm1\rangle$, also the state $|m_F = 0\rangle$ can be populated, because of, for example, magnetic fields in a direction not coinciding with the quantization axis. Hence, the atom–atom state effectively occupies a $3 \times 3$ state space. Assuming isotropic dephasing towards white noise, the fidelity relative to a maximally entangled state is therefore estimated as $\mathcal{F} \geq 1/9 + 8/9\overline{V}$, where $\overline{V}$ is average visibility in three orthogonal bases. The commonly used fidelity estimation from visibilities for an entangled two-qubit system is $\mathcal{F} \geq 1/4 + 3/4\overline{V}$. For the effective two-qutrit system, however, this would result in an higher fidelity and would overestimate the fidelity of the generated atom–atom state.

## Polarization control of long fibres

Stress or temperature induced polarization drifts in the long fibres are compensated using an automated polarization control. The polarization control is performed every 7 min, takes on average 20 s, and is based on a gradient descent optimization algorithm. In this way, polarization errors are kept below 1% during all measurements.

The fibre polarization is optimized using laser light at the single-photon frequency with sufficient optical power to be detected by conventional photodiodes. Two polarization directions are used, vertical and diagonal linear polarizations, which are alternated at 10 Hz. The light is overlapped at the nodes with the complete single-photon path up to the detectors. In both output arms of the beamsplitter, a flip-mirror reflects the classical light into a polarimeter during the optimization. Three fibre polarization controllers are connected to the fibre beamsplitter of the BSM: at both input ports and at one output port, and are set according to the result of the gradient descent optimization algorithm.

Polarization drifts in the long fibres are compensated in our setup, which includes a 700 metre fibre crossing public space and a four lane street. Recently, polarization drifts over a 10 km field deployed fibre were characterized and compensated[46]. In our setup with a configuration of 32.4 km spooled and 0.7 km field fibre, we observe similar drifts. This indicates that a setup including longer field deployed fibres does

not introduce substantially more drifts than observed now and that the currently used system can compensate for polarization drifts in longer field deployed fibres.

## Entanglement swapping fidelity

The single photons are detected with a BSM device consisting of a fibre beamsplitter, two polarizing beamsplitters (Wollaston Prisms) and four SNSPDs, as illustrated in Fig. 1 of the main text. In this setup, the fibre beamsplitter guarantees a unity spatial overlap of the photons originating from the nodes, whereas the polarizing beamsplitters and single-photon detectors allow for polarization analysis in both output ports. The detectors, labelled $H_1$, $V_1$, $H_2$ and $V_2$, are not photon number resolving, and hence we only distinguish between six coincidence combinations of two detectors, see Extended Data Table 2. For the purpose of a BSM, we can categorize these combinations into three groups: $D_+$, $D_-$, and $D_\varnothing$. Here, detector combinations in group $D_+$ and $D_-$ herald the Bell states $|\Psi^+\rangle$ and $|\Psi^-\rangle$, respectively, whereas combinations in group $D_\varnothing$ should not occur for perfectly interfering photons and are discarded in the analysis. However, the relative occurrence of these events is used in the following to quantify the two-photon interference contrast.

For not interfering photons, two-photon events are evenly distributed between the 16 possible detector combinations (not considering experimental imperfections). As the order of the detector combination is not of interest, for example, $(V_1,H_1)$ is similar to $(H_1,V_1)$, we end up with ten distinct coincidences and their probabilities, as listed in Extended Data Table 1. For perfectly interfering photons, the probabilities differ: the probability to detect the $D_\varnothing$ group vanishes and all four Bell states are detected with a probability of 1/4, whereby the $|\Phi^\pm\rangle$ Bell states fall into the group 'not detected' for the setup used.

The two-photon interference contrast is defined as[10]

$$C = 1 - \frac{2N_{D_\varnothing}}{N_{D_+} + N_{D_-}}, \tag{7}$$

where $N_k$ is the number of events in detection group $k$. With this definition and the probabilities of the different coincidences, the contrast equals zero for not interfering photons and one for perfectly interfering photons. See ref. [34] for a thorough analysis of the two-photon interference contrast and entanglement swapping fidelity, including experimental imperfections.

The interference contrast is measured as follows. During measurement runs all single-photon detection events are recorded, which allows to count the number of occurrences of the coincidence events for all three detection groups. Next, the interference contrast is evaluated using equation (7). To verify this method, we additionally evaluate the contrast of not interfering photons. This is done by analysing coincidence detections of two photons originating from distinct entanglement generation tries. In this way, the photons did not interfere because the photon wave-packets are completely separated in time. Fig. 2b of the main text shows exactly this for the $L = 6$ km measurement. Shown

are the normalized wrong coincidences, defined as $1-C$, for varying time differences between the photons ($\Delta\tau$). Note that the horizontal spacing of the measurement times equals the repetition rate of the entanglement generation tries.

The entanglement swapping fidelity is mainly limited by two effects. First, it is limited by experimental imperfections that reduce the indistinguishability of the two photons, for example, as discussed in the main text, by an imperfect time overlap of the two photon wave-packets. Second, it is limited by double excitations stemming from the finite duration of the excitation pulse. For a detailed description, see refs. [13,34].

## Data availability

The datasets generated during and/or analysed during the current study are available from the corresponding author upon request.

## Code availability

The code supporting the plots within this paper is available from the corresponding authors upon request.

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

**Acknowledgements** M.B. acknowledges the hospitality of the Ludwig-Maximilians-University of Munich group during several stays. We acknowledge helpful discussions with S. Kucera, K. Redeker and L. Knips. We acknowledge funding by the German Federal Ministry of Education and Research (Bundesministerium für Bildung und Forschung) within the projects Q.com.Q and Q.Link.X (contract nos. 16KIS0127, 16KIS0123, 16KIS0864 and 16KIS0880), the Deutsche Forschungsgemeinschaft (German Research Foundation) under Germany's Excellence Strategy – EXC-2111 – 390814868. W.Z. acknowledges funding by the Alexander von Humboldt foundation.

**Author contributions** M.B., C.B., T.v.L., W.Z., W.R. and H.W. devised the project. T.v.L., W.Z., F.F., S.E., Y.Z. and P.M. carried out the experiments. T.v.L. analysed the data. M.B. designed and constructed the QFC systems, assisted by T.B. R.G. designed and constructed the fluorescence collection setups in both nodes, assisted by M.S. W.Z., C.B. and H.W. jointly supervised the study. T.v.L. drafted the manuscript and all authors contributed to the final version. T.v.L., M.B. and F.F. contributed equally to this work.

**Funding** Open access funding provided by Universität des Saarlandes.

**Competing interests** The authors declare no competing interests.

**Additional information**
**Correspondence and requests for materials** should be addressed to Wei Zhang, Christoph Becher or Harald Weinfurter.

**a**

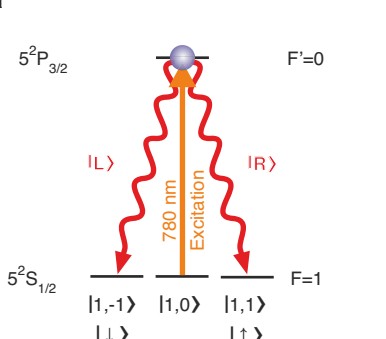
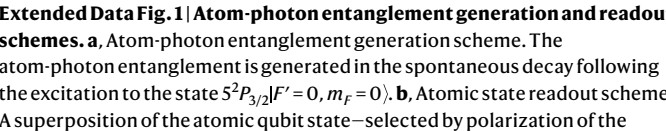

**b**

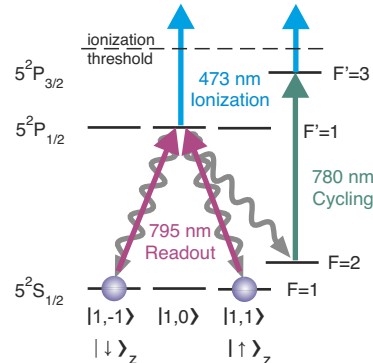

**Extended Data Fig. 1 | Atom-photon entanglement generation and readout schemes. a**, Atom-photon entanglement generation scheme. The atom-photon entanglement is generated in the spontaneous decay following the excitation to the state $5^2P_{3/2}|F' = 0, m_F = 0\rangle$. **b**, Atomic state readout scheme. A superposition of the atomic qubit state–selected by polarization of the readout light–is excited to the state $5^2P_{1/2}|F' = 1, m_{F'} = 0\rangle$ and subsequently ionized with a 473 nm laser pulse. If the atom decays to the state $5^2S_{1/2}|F = 2\rangle$ before it is ionized, a 780 nm laser pulse (labeled as Cycling) will transfer it to the state $5^2P_{3/2}|F' = 3\rangle$, which is ionized as well.

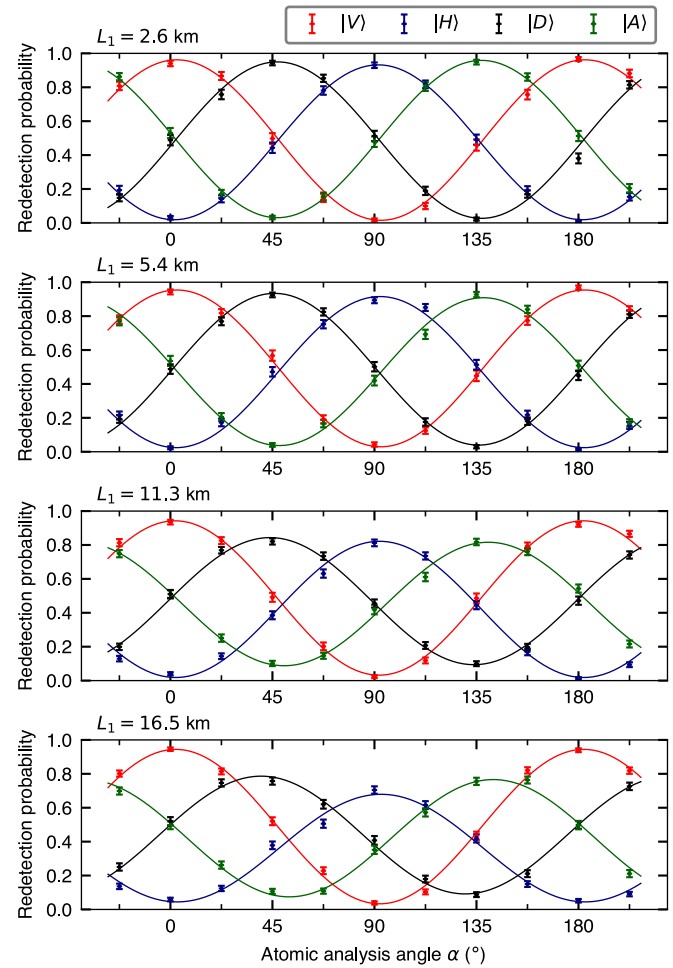

**Extended Data Fig. 2 | Atom-photon entanglement distribution for Node 1.**
The figues show the observed atom-photon state correlations for $L_1$ = 2.6, 5.4, 11.3, and 16.5 km, as indicated in the labels at the top left corner of each subfigure. The measurements include an atomic readout delay to allow for two-way communication with the middle station. The error bars indicate statistical errors of one standard deviation.

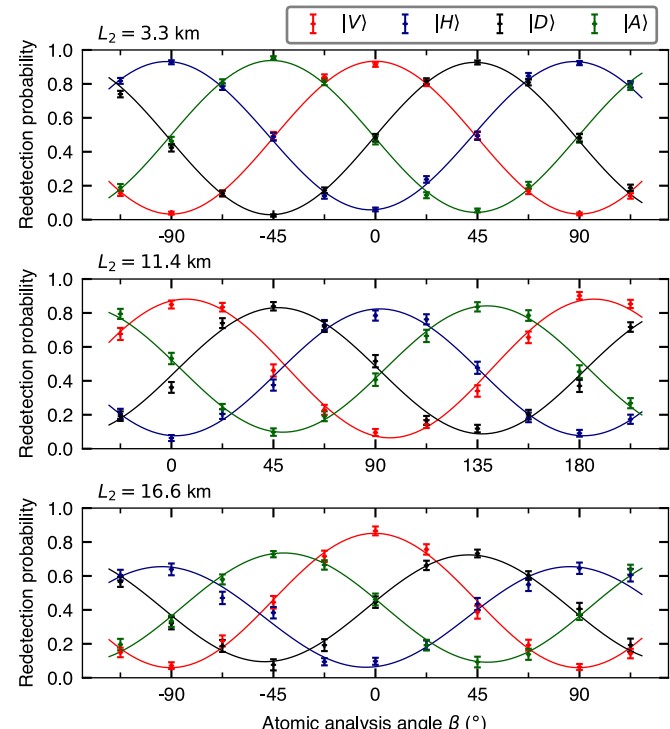

**Extended Data Fig. 3 | Atom-photon entanglement distribution for Node 2.** The figues show the observed atom-photon state correlations for $L_2$ = 3.3, 11.4, and 16.6 km, as indicated in the labels at the top left corner of each subfigure. The measurements include an atomic readout delay to allow for two-way communication with the middle station.

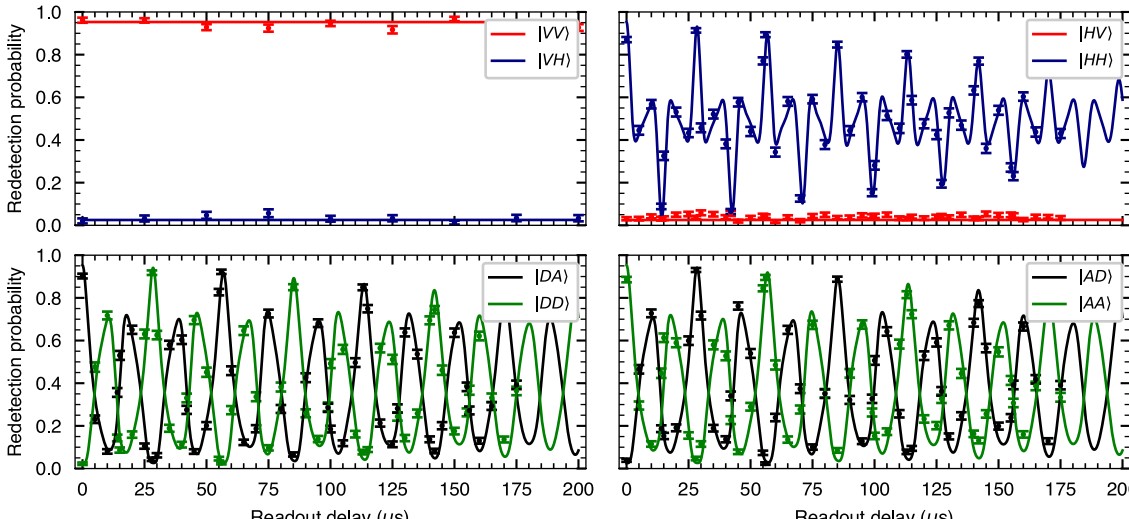

**Extended Data Fig. 4 | Atomic state evolution in two bases.** Atom-photon entanglement simulations (solid lines) and measurements (points) of Node 1 with photon detection at 780 nm using a 5 m long fibre. The atomic state readout orientation and time is varied to characterize the memory coherence. The labels in the legend indicate the |*atom photon*⟩ state analysed. Simulation parameters: trap waist 2.05 $\mu$m, trap depth 2.32 mK, atom temperature 50 $\mu$K, bias field $B_y$ = 75.5 mG, and field fluctuations $\Delta B_y$ = 0.5 mG Gaussian distributed. With these parameters, we observe de- and rephasing of the atomic state due to the longitudinal polarization components at the trap frequency of 70 kHz and a larmor precession at a frequency of 105 kHz.

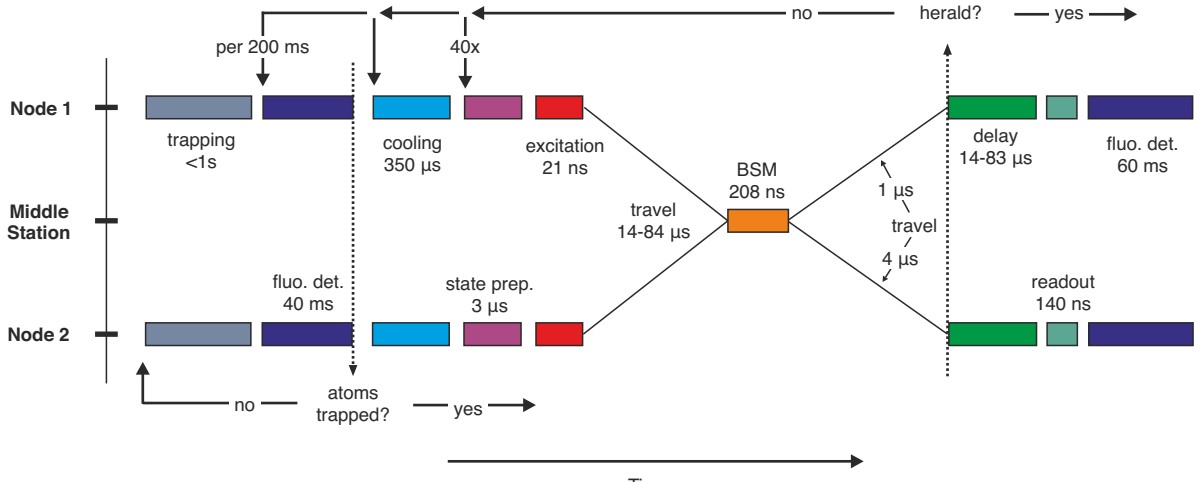

**Extended Data Fig. 5 | Experimental sequence of the atom-atom entanglement generation.** In both nodes a single atom is trapped and cooled using polarization gradient cooling. Next, the entanglement generation tries start containing state preparation and synchronous excitation of the atoms. The atoms are re-cooled after 40 unsuccessful entanglement generation tries. The travel time of the photons to the middle station equals 14 to 84 $\mu$s, depending on the fibre length. With regard to the assumption that the heralding signal is communicated back to the nodes along fibres of respective lengths, an additional delay is included to account for the communication time from the middle station back to the nodes. After a 200 ms interval of entanglement generation tries the presence of the atoms in the traps is checked using fluorescence collection with an APD (see text). The QFC takes place subsequent to the atomic excitation.

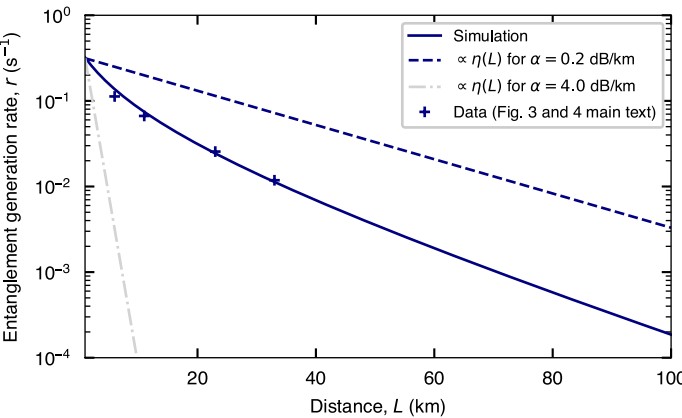

**Extended Data Fig. 6 | Entanglement generation rate for varying link lengths.** The plus markers show the observed entanglement generation rates ($r = \eta R$) for $L = 6, 11, 23,$ and 33 km, that is, the data presented in Fig. 3 and 4 of the main text. The blue solid line is a simulation of the entanglement generation rate based on Equations 4–6 and extrapolated up to $L = 100$ km. The dashed lines are proportional to the decay in success probability ($\propto \eta(L)$) for an attenuation rate in optical fibres of 0.2 dB/km at 1517 nm (blue dashed line) and 4.0 dB/km at 780 nm (gray dashed line). The difference between the blue solid and dashed lines originates from the reduced repetition rate $R(L)$ due to the longer communication times between the nodes and the middle station. For $L < 54$ km, the reduced repetition rate is the main contributor to the lower entanglement generation rate.

**Extended Data Table 1 | Long fibre configurations and corresponding atomic readout times**

| $L$ (km) | $L_1$ (km) | $L_2$ (km) | $A_1$ (dB) | $A_2$ (dB) | $t_1$ ($\mu$s) | $t_2$ ($\mu$s) |
|---|---|---|---|---|---|---|
| 6 | 2.6 | 3.3 | -0.7 | -0.8 | 28.5 | 35.5 |
| 11 | 5.4 | 5.5 | -1.5 | -1.3 | 57.1 | 71.0 |
| 23 | 11.3 | 11.4 | -3.3 | -2.8 | 114.2 | 124.3 |
| 33 | 16.5 | 16.6 | -4.5 | -4.1 | 171.2 | 177.5 |

The fibre link lengths and corresponding atomic readout times for the experiments presented in the main text and the atom-photon characterization measurements. $L_1$ ($L_2$) equals the fibre length between Node 1 (Node 2) and the middle station. $A_1$ ($A_2$) gives the attenuation in the fibre network between the node and the middle station, this includes inefficiencies of fibre-to-fibre connectors.

**Extended Data Table 2 | Possible two-photon coincidences**

| detection | coincidence | no interference | perfect interference |
|---|---|---|---|
| not detected | $H_1,H_1$ | 1/16 | 1/8 |
| | $H_2,H_2$ | 1/16 | 1/8 |
| | $V_1,V_1$ | 1/16 | 1/8 |
| | $V_2,V_2$ | 1/16 | 1/8 |
| $D_\varnothing \rightarrow$ discarded | $H_1,H_2$ | 1/8 | 0 |
| | $V_1,V_2$ | 1/8 | 0 |
| $D_+ \rightarrow |\Psi^+\rangle$ | $H_1,V_1$ | 1/8 | 1/8 |
| | $H_2,V_2$ | 1/8 | 1/8 |
| $D_- \rightarrow |\Psi^-\rangle$ | $H_1,V_2$ | 1/8 | 1/8 |
| | $V_1,H_2$ | 1/8 | 1/8 |

The probabilities per coincidence event are given for not interfering photons and perfectly interfering photons.