## [Peer Review File · Nature]

Manuscript Title: Entangling single atoms over 33 km telecom fibre

Reviewer Comments & Author Rebuttals

Reviewer Reports on the Initial Version:

Referees' comments:

Referee #1 (Remarks to the Author):

van Leent et al present implementation of entanglement between two remote atoms over telecom fibers. To achieve this, photons emitted by atoms are converted to telecom, for low-loss fiber propagation. The experiment builds on their prior advances in the system. The main new element is the incorporation of telecom conversion.

I find the study to be an impressive achievement. The data are extensive, and are appropriately analyzed. The manuscript is appropriate for Nature, the introduction and conclusion in particular are clear and well written. I have a (cosmetic) suggestion for the authors to consider: in the abstract only the fibers length are mentioned, whereas it would be helpful for a casual reader to ascertain quickly that the physical separation between the two atoms was 800 m.

Referee #2 (Remarks to the Author):

The manuscript reports a beautiful experiment demonstrating entanglement between two single trapped atoms placed in independent apparatuses separated by a physical distance of 400 m and by up to 33 km of optical fibers. The authors first create entanglement between each atom and a photon, which is then converted to telecommunication wavelengths. The two telecom photons from the two nodes are then mixed at a beam splitter at a central location, where a Bell state measurement is performed. This measurement heralds entanglement between the two atoms.

To verify this entanglement, the state of the two atoms is measured after a storage time longer than the two-way communication time between the atoms and the central station. In that configuration, the entanglement is still stored in the atoms when the result of the Bell state measurement arrives. With a separation of 6 km, the authors could show a violation of the CHSH inequality with the two atoms. Then, they measure the fidelity of the atom-atom entangled state as a function of fiber distance, and could obtain a fidelity of 62 % over 33 km of optical fiber.

In addition, the authors then convincingly verify that the loss of fidelity when increasing the distance is primarily due to the limited coherence time of the atoms, and that no significant decoherence is induced by the transmission of the photons in the long fibers.

The paper is clearly written and the experimental results are sound and very convincing.

The fast generation of high-fidelity heralded entanglement between remote quantum nodes is a key capability for building quantum networks. This experiment of van Leent et al represents a significant milestone in that direction. Similar experiments have been performed at short distances e.g. with trapped ions (e.g. Nature Physics 11, 37 (2015)) or single atoms (Science, 337, 72, 2012), or at km distance using NV centers (Nature 526, 682, 2015). Increasing the distance to tens of km is significantly more challenging because it requires to use photons at telecom wavelengths, to achieve sufficient storage time in the single atoms and to ensure that the transmission of the photons in the long fibers do not lead to decoherence (by e.g. polarization fluctuations). Related results have been also achieved recently with cold atomic ensembles (Nature 578, 240, 2020), however in that experiment, the storage time in the atoms was not sufficient to allow two-way communication to the middle station. In other words, the memories had to be read-out before the heralding signal from the BSM came back to the memory. The current manuscript reports for the first time to my knowledge an experiment where the entanglement is stored for long enough in the atoms to demonstrate heralded the presence of matter-matter entanglement stored in the atoms, over tens of km of optical fiber. This is a crucial step to scale up quantum network.

The work reported in this paper represents a very impressive experimental demonstration and a very significant advance towards the realization of large-scale quantum networks. I support publication in Nature, but I would like the authors to address the following points before publication.

- In the bold paragraph, they authors write: “However, despite vast efforts, long-distance fibre based network links have not been realized yet “. The authors should in my opinion be more careful with this claim. Network links do not necessarily need to involve matter systems, and there are many experiments that demonstrated entangled photon distribution over long distances. Also, the experiment reported by Pan’s group in 2020 could be counted as long distance fiber network link. What the authors have demonstrated is the first long distance fibre-based quantum link with heralded entanglement between matter nodes.

- In the introduction, the author say that “it is necessary to convert light to telecom wavelength”. I agree that the transmitted photon should be at telecom wavelengths, but there are other ways to achieve that, e.g. by using non-degenerate photon pair sources, as for example in ref 25.

- It would be useful for the reader to explicitly mention the photon collection efficiency from each node in the main text.

- The authors show that the main cause for the loss of fidelity of the entanglement as a function of distance is the atomic decoherence. In the conclusion, they conjecture that an improvement of coherence time would enable entanglement over 100 km with a fidelity of 80 %. They should mention explicitly what should be the memory coherence time to achieve such an experiment.

- While the experiment is very impressive, the entanglement rate at 33 km is quite modest (1 event/208 s). The

authors should infer the expected entanglement rate at longer distances (e.g. at the 100 km they mention in the conclusion) which I expect would be very small, and discuss if potential improvement could lead to more practical rates (e.g. use optical cavities to increase the collection efficiency).

- In relation to the previous point, in order to efficiently scale up quantum networks with multiple links and nodes, it is important to reach a quantum link efficiency larger than 1, i.e to create entanglement faster than it is lost. This is a very challenging goal that was reached over short distances with trapped ions (Nature Physics 11, 37 (2015)) and NV centers (Nature 558, 268 (2018)), but so far never at long distances. This regime is also not reached in the present experiment. The authors should estimate the current quantum link efficiency (heralding rate/entanglement decay rate) and discuss

Referee #3 (Remarks to the Author):

In the manuscript “Entangling single atoms over 33km telecom fibre”, authors demonstrate generation of heralded entanglement between two single atoms at two separate buildings. Most electronic transitions are not at telecommunication wavelength. Authors show that photons extracted from the single atoms at 780nm can be efficiently converted to 1517nm for low loss propagation over telecom fibre up to total 33kms of fibre length. In order to maintain the spin-polarization entanglement between the single atom and the emitted photon, the authors implement a polarization-preserving quantum frequency conversion and an active polarization drift compensation through the long fiber. The quantum frequency conversion does not introduce excessive noise and is highly efficient with external efficiency of 57%. These are all essential elements for a future quantum network. Even though, photon-atom entanglement and quantum frequency conversion have been performed in the past, heralded atom-atom entanglement through telecommunication fibre has not been achieved before this work (with the exception of this paper that recently appeared on arxiv at arXiv:2201.11953v). Despite the importance of this manuscript by van Leent et al., I have the following criticism that is preventing me from recommending the paper for publication in Nature.

Major comments:

- 1) First, the experiment has not been performed over deployed fibre. This can introduce additional challenges regarding preserving the polarization state of the photons.
- 2) Second, the polarization-dependent ionizing measurement is only useful for confirmation of the entanglement. Authors have demonstrate a viable path that such atom-atom entanglement can be extended beyond one link.
- 3) Third, the quality of the experiment is high enough to demonstrate violation of the CHSH inequality. However, authors don't discuss if the fidelity is high enough to reasonably expect entanglement distillation based on this system and if not what are the required improvements to get there?
- 4)The ionizing measurement scheme can be very limiting when one considers implementation beyond a single link. I would like to know if the authors have an alternative scheme to present that would alleviate such a limitation.

5) Based these points, I am not convinced that this demonstration (which is an important milestone) is the effective solution to reach a quantum network beyond one link. To clarify, it is unclear whether other architectures can be more efficient. For example, single atoms as an emissive memory can be a major restriction when one considers implementing more than one link. I recommend the authors to compare the best possible performance of their system with other architectures. For example, combination of absorptive memories with non-degenerate sources of photons (where telecom photon is used for propagation and the other photon is mapped on to the memory).

Comments on the presentation:

5) There are important elements of this work that is not being discussed in the manuscript. Most importantly, the single atom level structure and how it's used in generating the atom-photon entanglement. In addition, authors don't offer any details on the ionizing measurement. Ref. 21 by the same group has more details on these aspects but the paper should be self-sufficient and the reader should not have to rely on other references to understand the details. Fig 1 can be modified to contain some these details.

Minor technical questions:

6) I don't understand the re-defined fidelity threshold. Authors should be able to offer a full state tomography and a direct comparison to an entangled state.

7) It would be helpful to understand the change in the performance (in terms of rate and fidelity) with respect to increasing or decreasing the acceptance window for the two-photon detection. I encourage the authors to comment on this in their revised manuscript.

Author Rebuttals to Initial Comments:

PREAMBLE

We like to thank the reviewers for their inputs and constructive comments. Below, we provide a point-to-point response to all of the comments and questions, as well as a summary of changes to the original version of the manuscript.

REFEREE #1

A. Remarks to the Author

Van Leent et al present implementation of entanglement between two remote atoms over telecom fibers. To achieve this, photons emitted by atoms are converted to telecom, for low-loss fiber propagation. The experiment builds on their prior advances in the system. The main new element is the incorporation of telecom conversion.

I find the study to be an impressive achievement. The data are extensive, and are appropriately analyzed. The manuscript is appropriate for Nature, the introduction and conclusion in particular are clear and well written. I have a (cosmetic) suggestion for the authors to consider: in the abstract only the fibers length are mentioned, whereas it would be helpful for a casual reader to ascertain quickly that the physical separation between the two atoms was 800 m.

B. Response to the Referee

We thank referee #1 for the positive comments on our work and the recommendation for publication in Nature.

We adopted the suggestions to more prominently indicate the physical separation between the two atoms and hence added this distance to the abstract.

REFEREE #2

A. Remarks to the Author

The manuscript reports a beautiful experiment demonstrating entanglement between two single trapped atoms placed in independent apparatuses separated by a physical distance of 400 m and by up to 33 km of optical fibers. The authors first create entanglement between each atom and a photon, which is then converted to telecommunication wavelengths. The two telecom photons from the two nodes are then mixed at a beam splitter at a central location, where a Bell state measurement is performed. This measurement heralds entanglement between the two atoms.

To verify this entanglement, the state of the two atoms is measured after a storage time longer than the two-way communication time between the atoms and the central station. In that configuration, the entanglement is still stored in the atoms when the result of the Bell state measurement arrives. With a separation of 6 km, the authors could show a violation of the CHSH inequality with the two atoms. Then, they measure the fidelity of the

atom-atom entangled state as a function of fiber distance, and could obtain a fidelity of 62 % over 33 km of optical fiber.

In addition, the authors then convincingly verify that the loss of fidelity when increasing the distance is primarily due to the limited coherence time of the atoms, and that no significant decoherence is induced by the transmission of the photons in the long fibers.

The paper is clearly written and the experimental results are sound and very convincing.

The fast generation of high-fidelity heralded entanglement between remote quantum nodes is a key capability for building quantum networks. This experiment of van Leent et al represents a significant milestone in that

direction. Similar experiments have been performed at short distances e.g. with trapped ions (e.g. Nature Physics 11, 37 (2015)) or single atoms (Science, 337, 72, 2012), or at km distance using NV centers (Nature 526, 682, 2015). Increasing the distance to tens of km is significantly more challenging because it requires to use photons at telecom wavelengths, to achieve sufficient storage time in the single atoms and to ensure that the transmission of the photons in the long fibers do not lead to decoherence (by e.g. polarization fluctuations). Related results have been also achieved recently with cold atomic ensembles (Nature 578, 240, 2020), however in that experiment, the storage time in the atoms was not sufficient to allow two-way communication to the middle station. In other words, the memories had to be read-out before the heralding signal from the BSM came back to the memory. The current manuscript reports for the first time to my knowledge an experiment where the entanglement is stored for long enough in the atoms to demonstrate heralded the presence of matter-matter entanglement stored in the atoms, over tens of km of optical fiber. This is a crucial step to scale up quantum network.

The work reported in this paper represents a very impressive experimental demonstration and a very significant advance towards the realization of large-scale quantum networks. I support publication in Nature, but I would like the authors to address the following points before publication.

B. Response to the Referee

We thank referee #2 for the detailed and constructive comments on our work, the recognition of the significance of the obtained results, and the support for publication in Nature.

Point-to-point response:

- 1. Comment:** In the bold paragraph, they authors write: "However, despite vast efforts, long-distance fibre based network links have not been realized yet." The authors should in my opinion be more careful with this claim. Network links do not necessarily need to involve matter systems, and there are many experiments that demonstrated entangled photon distribution over long distances. Also, the experiment reported by Pan's group in 2020 could be counted as long distance fiber network link. What the authors have demonstrated is the first long distance fibre-based quantum link with heralded entanglement between matter nodes.

Response: We should indeed have been more careful here; we have revised the abstract, taken out this particular sentence, and now emphasize the herald and mention the quantum memories.

2. **Comment:** In the introduction, the author say that: "it is necessary to convert light to telecom wavelength." I agree that the transmitted photon should be at telecom wavelengths, but there are other ways to achieve that, e.g. by using non-degenerate photon pair sources, as for example in ref 25.

Response: We agree with this comment and, accordingly, changed the sentence to: "*.. it is necessary to operate at telecom wavelengths. Employing quantum frequency conversion [16-20], light-matter entanglement distribution at the low loss telecom band has recently been demonstrated..*"

3. **Comment:** It would be useful for the reader to explicitly mention the photon collection efficiency from each node in the main text.

Response: The photon collection efficiencies of the nodes are added to the Section "Quantum network link with telecom interfaces" as follows: "*A custom made high-NA objective is used to collect the atomic fluorescence into single-mode fibre with an efficiency of 1.0% (1.1%) after an excitation attempt in Node 1 (Node 2).*"

4. **Comment:** The authors show that the main cause for the loss of fidelity of the entanglement as a function of distance is the atomic decoherence. In the conclusion, they conjecture that an improvement of coherence time would enable entanglement over 100 km with a fidelity of 80%. They should mention explicitly what should be the memory coherence time to achieve such an experiment.

Response: A 100 km link requires a good readout contrast at 500 us coherence time (two-way communication time to the middle station in a symmetric setup). To guarantee that, a 10 times longer coherence time (T2), i.e., 5 ms, is necessary.

Recently, we realized a new encoding of our qubit, involving states that are a factor 500 less sensitive to magnetic field fluctuations (using the methods presented in Nature Photonics, 12, 18 (2018)). This increased the memory coherence time to about 6 ms and hence should enable for entanglement generation over 100 km fibre links without significant memory decoherence. We are implementing a spin echo, with which we expect to increase the coherence time by another order.

To emphasise this and explicitly mention the required coherence time in the manuscript, we added the following sentence to the Section "Discussion and outlook": "*.. the coherence time of the atomic memories can be increased by the implementation of first a new trap geometry to mitigate the position-dependent de-phasing and second a state-transfer to a qubit encoding less sensitive to magnetic fields [Nature Photonics, 12, 18 (2018)] making possible coherence times in excess of 5 ms, and thus distances of 100 km without significant memory decoherence.*"

5. **Comment:** While the experiment is very impressive, the entanglement rate at 33 km is quite modest (1 event/208 s). The authors should infer the expected entanglement rate at longer distances (e.g. at the 100 km they mention in the conclusion) which I expect would be very small, and discuss if potential improvement could lead to more practical rates (e.g. use optical cavities to increase the collection efficiency).

Response: Please see the response to Comment 6 below.

The expected (and observed) entanglement generation rates for link lengths up to 100 km are presented in the additional Methods section "Entanglement generation rate".

6. **Comment:** In relation to the previous point, in order to efficiently scale up quantum networks with multiple links and nodes, it is important to reach a quantum link efficiency larger than 1, i.e. to create entanglement faster than it is lost. This is a very challenging goal that was reached over short distances with trapped ions (Nature Physics 11, 37 (2015)) and NV centers (Nature 558, 268 (2018)), but so far never at long distances. This regime is also not reached in the present experiment. The authors should estimate the current quantum link efficiency (heralding rate/entanglement decay rate) and discuss.

Response: For the realization of quantum repeater protocols, it is, indeed, necessary to have a quantum link efficiency larger than one. This efficiency depends on the link length and, for our apparatus, equals to $330\mu\text{s}/9\text{s} = 3.7 \cdot 10^{-5}$ for $L = 6$ km and $330\mu\text{s}/85\text{s} = 3.9 \cdot 10^{-6}$ for $L = 33$ km.

Clearly, improvements are needed to achieve a link efficiency above one. For improvements on the coherence time we refer to our response on Comment 4 above; to improve the entanglement generation rate there are various strategies for neutral single atoms. For one, optical cavities can greatly enhance the collection efficiency of atoms, for single rubidium atoms, photon collection efficiencies of 56%, including coupling into a single mode fibre, have been reported (Nature Physics 16, 647 (2020)). Another, compatible, approach is to increase the event rate by leveraging the scalability of the single atom platform and implement additional dipole traps (e.g., Science 354, 6315 (2016), Science 373, 6562(2021)), allowing to multiplex the entanglement generation process. A third, more complex, but efficient, approach is adding a non-destructive measurement device for the photon (Nature 591, 570 (2021)). This would herald the successful collection of a photon after which it can be sent to other nodes via long fibres.

Combining these approaches, however being a complex task, can improve the quantum link efficiency by more than 6 orders of magnitude. State-of-the-art demonstrations show improvements of 3 orders of magnitude in coherence time, 2 orders of magnitude in entanglement generation efficiency (squared

dependence on photon collection efficiency for two-photon interference), and >2 orders of magnitude for a scale up number of traps.

To discuss the quantum link efficiency and possible improvements in the manuscript, we revised the Section "Discussion and outlook" as follows:

The results clearly indicate the feasibility of turning to large-scale quantum networks and of increasing the line-of-sight separation of the nodes to tens of kilometres by employing efficient quantum frequency conversion. In order to evaluate the performance of entanglement generation in future quantum networks the so called quantum link efficiency was introduced recently [Nature 558, 268 (2018)]. It is defined as the ratio of the entanglement generation rate over the entanglement decay rate and describes how efficiently one can use entanglement as a resource in future quantum networks. Ideally, it should exceed one, i.e., entanglement is available on demand as it is generated faster than it decays. However, the link efficiency decays rapidly with length due to both the exponential decrease of the signal detection probability, but, even more dominantly for the link lengths realised here, also due to the waiting times for classical communication between the nodes (see Methods). In this proof-of-principle demonstration the link efficiency was at the 10^{-5} to 10^{-6} level, mainly due to the low photon collection efficiency, relatively short coherence times, and long fibre links.

To raise the link efficiency a number of improvements are feasible: the coherence time of the atomic memories can be increased by the implementation of first a new trap geometry to mitigate the position-dependent dephasing and second a state-transfer to a qubit encoding less sensitive to magnetic fields [Nature Photonics, 12, 18 (2018)] making possible coherence times in excess of 5 ms, and thus distances of 100 km without significant memory decoherence. On the entanglement generation side, optical cavities

could enhance the fluorescence collection efficiency into fibres. For single rubidium atoms, photon collection efficiencies up to 11% have been demonstrated [Nature Physics 16, 647 (2020)]. Yet, even for an almost ideal experimental platform the entangling rate cannot be increased arbitrarily high. To further increase the event rate on the long run it will be mandatory to parallelize the entanglement distribution in order to regain the scalability. For the neutral atom platform this could be achieved by realising atom trap arrays [Science 373, 6562 (2021)] and hence potentially increase the entanglement generation rate by orders of magnitude. This concept will also lay the basis to realise quantum repeater network nodes performing entanglement purification and Bell-state measurements via local atom-atom gate operations in the array, e.g., employing Rydberg interactions [Reviews of Modern Physics, 82, 2313 (2010)].

We hope that with this reply and the revisions in the manuscript Referee #2 will find our manuscript suitable for publication in Nature.

REFEREE #3

A. Remarks to the Author

In the manuscript "Entangling single atoms over 33km telecom fibre", authors demonstrate generation of heralded entanglement between two single atoms at two separate buildings. Most electronic transitions are not at telecommunication wavelength. Authors show that photons extracted from the single atoms at 780nm can be efficiently converted to 1517nm for low loss propagation over telecom fibre up to total 33kms of fibre length. In order to maintain the spin-polarization entanglement between the single atom and the emitted photon, the authors implement a polarization-preserving quantum frequency conversion and an active polarization drift compensation through the long fiber. The quantum frequency conversion does not introduce excessive noise and is highly efficient with external efficiency of 57%. These are all essential elements for a future quantum network. Even though, photon-atom entanglement and quantum frequency conversion have been performed in the past, heralded atom-atom entanglement through telecommunication fibre has not been achieved before this work (with the exception of this paper that recently appeared on arxiv at arXiv:2201.11953v). Despite the importance of

this manuscript by van Leent et al., I have the following criticism that is preventing me from recommending the paper for publication in Nature.

B. Response to the Referee

We thank Referee #3 for the constructive comments on our work and recognition of the importance of the obtained results.

Point-to-point response:

Major comments:

1. Comment: First, the experiment has not been performed over deployed fibre. This can introduce additional challenges regarding preserving the polarization state of the photons.

Response: We agree that entanglement generation over field deployed fibres do, indeed, introduce additional technical challenges—which should not be underestimated. For our setup, including a 700 metre fibre crossing public space and a four lane street, we successfully compensate for polarization drifts with an automated optimization system. Recently, researchers at NUS in Singapore characterized and compensated polarization state drifts in a 10 km field deployed fibre (Optics Express 29, 23, 37075-37080 (2021)); in our setup—in a configuration with 32.4 km spooled and 0.7 km field fibres—we see similar polarization drifts as they observed. This gives us confidence that a setup including longer field deployed fibres does not introduce significantly more polarization drifts than we observe now and that the currently used system could also compensate long field deployed fibres.

To elaborate on this in the manuscript, we have included a summary of the above discussion in the Methods section "Polarization control of long fibers".

2. **Comment:** Second, the polarization-dependent ionizing measurement is only useful for confirmation of the entanglement. Authors have demonstrate a viable path that such atom-atom entanglement can be extended beyond one link.

Response: The extension of entanglement generation beyond one link, i.e., a quantum repeater, is the key building block for a quantum network. We fully agree that this is of great interest and it should be possible to realize with a potential platform. A possible approach based on our single-atom trap setups would exploit the scalability of the number of atoms traps. It involves the trapping of another single-atom at a distance of $\sim 10\mu\text{m}$, e.g., via an array of optical tweezers, and use Rydberg interactions between the atoms. For example, we store two atoms in Node A, and individually entangle these atoms with an atom in Node B and with an atom in Node C. After achieving this, the atoms in the distant Nodes B and C can be entangled by a Bell-state measurement on the two atoms in Node A via local atom-atom gate operations.

To elaborate on this in the manuscript, we added the following parts to the "Discussion and outlook" section of the paper: *".. For the neutral atom platform this could be achieved by realising atom trap arrays [Science 373, 6562 (2021)] and hence potentially increase the entanglement generation rate by orders of magnitude. This concept will also lay the basis to realise quantum repeater network nodes performing entanglement purification and Bell-state measurements via local atom-atom gate operations in the array, e.g., employing Rydberg interactions [Reviews of Modern Physics, 82, 2313 (2010)]."*

The method described above can be realized with a destructive readout of the atomic states. The possible need to reload an atom after a readout event, however, will effect the entanglement generation rate—how much limitation occurs will depend on the implemented protocol. In the domain of neutral atoms, trap

losses and their replenishment are common and well under control, e.g., demonstrated by the deterministic loading of (2D) arrays of single atoms (Science 354, 6315 (2016)). However, while being common, trap losses and destructive readout schemes are not fundamental for neutral atoms. By achieving lower pressures in the trap chamber and increasing photon collection efficiencies, atom lifetimes of more than 7 minutes and non-destructive, 2000-times repeated readout schemes have been demonstrated (Phys. Rev. Lett. 122, 173201 (2019)).

7. **Comment:** Third, the quality of the experiment is high enough to demonstrate violation of the CHSH inequality. However, authors don't discuss if the fidelity is high enough to reasonably expect entanglement distillation based on this system and if not what are the required improvements to get there?

Response: For the achieved fidelities entanglement distillation is possible; this requires a minimum state fidelity of 50%. It also requires, however, another pair of entangled atoms, as discussed above. Since this goes beyond the scope of this paper, we decided to only briefly mention this in the Discussion and outlook section of the manuscript.

8. **Comment:** The ionizing measurement scheme can be very limiting when one considers implementation beyond a single link. I would like to know if the authors have an alternative scheme to present that would alleviate such a limitation.

Response: Please see the response to Comment 2 above.

9. **Comment:** Based these points, I am not convinced that this demonstration (which is an important milestone) is the effective solution to reach a quantum network beyond one link. To clarify, it is unclear whether other architectures can be more efficient. For example, single atoms as an emissive memory can be a major restriction when one considers implementing more than one link. I recommend the authors to compare the best possible performance of their system with other architectures. For example, combination of absorptive memories with non-degenerate sources of photons (where telecom photon is used for propagation and the other photon is mapped on to the memory).

Response: Currently, a lot of effort is made to study various platforms and architectures to serve as a quantum network node, and, indeed, there is no platform, yet, with a significant advantage to all others. For single neutral atoms, many capabilities needed to serve as quantum network node have been demonstrated over the last decades. For example, atom-atom gates, entanglement distribution at telecom wavelengths, long storage times, and scalability of the number of atom traps. For a recent review we refer to (Adv. Quantum Technol., 3, 1900141 (2020)). We are convinced that neutral atoms still belong to the viable systems.

Comments on the presentation:

10. **Comment:** There are important elements of this work that is not being discussed in the manuscript. Most importantly, the single atom level structure and how it's used in generating the atom-photon entanglement. In addition, authors don't offer any details on the ionizing measurement. Ref. 21 by the same group has more details on these aspects but the paper should be self-sufficient and the reader should not have to rely on other references to understand the details. Fig 1 can be modified to contain some these details.

Response: We agree that the main results of the paper should be self-sufficient. We therefore included schematics and a detailed description of the atom-photon entanglement generation and the atomic state readout schemes. Due to limited space, we decided to add the figures and description to the Methods section of the manuscript.

Minor technical questions:

3. Comment: I don't understand the re-defined fidelity threshold. Authors should be able to offer a full state tomography and a direct comparison to an entangled state.

Response: We think that the concern is about the estimation of the fidelity of the atom-atom state as $F \geq 1/9 + 8/9\bar{V}$, where \bar{V} equals the observed average visibility in three bases.

We added the following paragraph to the Methods section to elaborate on this:

"Atom-atom state fidelity estimation"

The state of the atomic quantum memories is encoded in two magnetic sublevels of the rubidium ground state $5S_{1/2} |F = 1\rangle$, which, however, is a spin-1 system. Besides the qubit states $|mF = \pm 1\rangle$, also the state $|mF = 0\rangle$ can be populated, due to, for example, magnetic fields in a direction not coinciding with the quantization axis. Hence, the atom-atom state effectively occupies a 3×3 state space. Assuming isotropic dephasing towards white noise, the fidelity relative to a maximally entangled state is therefore estimated as $F \geq 1/9 + 8/9\bar{V}$, where \bar{V} is average visibility in three orthogonal bases. The commonly used fidelity estimation from visibilities for an entangled 2-qubit system is $F \geq 1/4 + 3/4\bar{V}$. For the effective qutrit, however, this would result in an higher fidelity and would overestimate the fidelity of the generated atom-atom state."

By a tomography measurement of the atom-atom state we confirmed this fidelity estimation in an experiment without frequency conversion (not yet published).

4. Comment: It would be helpful to understand the change in the performance (in terms of rate and fidelity) with respect to increasing or decreasing the acceptance window for the two-photon detection. I encourage the authors to comment on this in their revised manuscript.

Response: There are two effects that reduce the system's performance that depend on the duration (and timing) of the acceptance window. First, the background introduced by the two quantum frequency conversion devices, whereby the signal to background ratio is best for a short acceptance window around the peak of the photon detection probability, see Figure 2a of the manuscript. Second, the quality of the two-photon interference, which is mainly limited by double excitation events in one node. Not accepting early detection events effectively filters out these events. This effect requires a thorough analysis and therefore we refer to our recent paper (arXiv:2110.00575, Reference 9 of the revised manuscript) for further details.

For the $L = 6$ km fibre configuration, accepting all coincidences in the hard-wired 208 ns long photon window results in an interference contrast of 0.895(8) and a signal-to-background ratio (SBR) of 22; resulting in an observed average atom-atom state fidelity of 0.772(10). When applying the 70 ns photon window, as presented in the manuscript, we observe an increase in interference contrast to 0.955(7) and a SBR to 48; resulting in an average atom-atom state fidelity of 0.816(13). Note that the interference contrast is not corrected for dark counts. The rate is directly proportional to the ratio of accepted coincidence events, in this case, the 70 ns window has a rate equal to 65% of the 208 ns acceptance window.

To comment on this in the manuscript, we added the following sentence to the Section "Entanglement distribution at telecom wavelength": *"(When also accepting the remaining 35% of the coincidences in the hard-wired acceptance window, the average observed fidelity for the states $|\Psi^\pm\rangle$ equals 0.772(10), with a SBR of 22, and an interference contrast of 0.895(8).)"*

We hope that with this reply and the revisions in the manuscript Referee #3 will find our manuscript suitable for publication in Nature.